# Direct Infusion Based Metabolomics Identifies Metabolic Disease in Patients’ Dried Blood Spots and Plasma

**DOI:** 10.3390/metabo9010012

**Published:** 2019-01-11

**Authors:** Hanneke A. Haijes, Marcel Willemsen, Maria van der Ham, Johan Gerrits, Mia L. Pras-Raves, Hubertus C. M. T. Prinsen, Peter M. van Hasselt, Monique G. M. de Sain-van der Velden, Nanda M. Verhoeven-Duif, Judith J. M. Jans

**Affiliations:** 1Section Metabolic Diagnostics, Department of Genetics, University Medical Centre Utrecht, Utrecht University, Lundlaan 6, 3584 EA Utrecht, The Netherlands; a.m.willemsen.marcel@gmail.com (M.W.); M.vanderHam-3@umcutrecht.nl (M.v.d.H.); J.Gerrits@umcutrecht.nl (J.G.); m.pras@amc.uva.nl (M.L.P.-R.); B.Prinsen@umcutrecht.nl (H.C.M.T.P.); M.G.deSain@umcutrecht.nl (M.G.M.d.S.-v.d.V.); N.Verhoeven@umcutrecht.nl (N.M.V.-D.); 2Section Metabolic Diseases, Department of Child Health, Wilhelmina Children’s Hospital, University Medical Centre Utrecht, Utrecht University, Lundlaan 6, 3584 EA Utrecht, The Netherlands; P.vanHasselt@umcutrecht.nl; 3Department of Clinical Epidemiology and Biostatistics, Amsterdam University Medical Centre, Meibergdreef 9, 1105 AZ Amsterdam, The Netherlands

**Keywords:** metabolomics, inborn errors of metabolism, direct-infusion mass spectrometry, IEM, DIMS

## Abstract

In metabolic diagnostics, there is an emerging need for a comprehensive test to acquire a complete view of metabolite status. Here, we describe a non-quantitative direct-infusion high-resolution mass spectrometry (DI-HRMS) based metabolomics method and evaluate the method for both dried blood spots (DBS) and plasma. 110 DBS of 42 patients harboring 23 different inborn errors of metabolism (IEM) and 86 plasma samples of 38 patients harboring 21 different IEM were analyzed using DI-HRMS. A peak calling pipeline developed in R programming language provided Z-scores for ~1875 mass peaks corresponding to ~3835 metabolite annotations (including isomers) per sample. Based on metabolite Z-scores, patients were assigned a ‘most probable diagnosis’ by an investigator blinded for the known diagnoses of the patients. Based on DBS sample analysis, 37/42 of the patients, corresponding to 22/23 IEM, could be correctly assigned a ‘most probable diagnosis’. Plasma sample analysis, resulted in a correct ‘most probable diagnosis’ in 32/38 of the patients, corresponding to 19/21 IEM. The added clinical value of the method was illustrated by a case wherein DI-HRMS metabolomics aided interpretation of a variant of unknown significance (VUS) identified by whole-exome sequencing. In summary, non-quantitative DI-HRMS metabolomics in DBS and plasma is a very consistent, high-throughput and nonselective method for investigating the metabolome in genetic disease.

## 1. Introduction

With unprecedented pace, the number of known inborn errors of metabolism (IEM) is expanding. It is a challenge to diagnose this diverse spectrum of diseases in a timely manner. Thus, there is an emerging need for a comprehensive test to acquire a complete view of metabolite status [1]. To meet this need, metabolomics—the parallel determination of thousands of small-molecule metabolites—is regarded as the way forward. Using metabolomics, all intermediates and final products of metabolic pathways in the body can potentially be measured. Metabolomics with exact quantification of a subset of predefined metabolites has been used for small-scale diagnostic screening [2,3,4] and for biomarker identification in predefined patient groups [5]. While broad quantitative metabolomics assays are already a big step forward compared to the focused targeted assays currently used in metabolic diagnostics, semi- and non-quantitative metabolomics can potentially detect an even larger range of metabolites. Non-quantitative metabolomics has mainly been performed in toxicology, neurology, cardiovascular, and cancer research. However, also in the IEM field, both combined approaches of quantitative and non-quantitative metabolomics [6,7,8,9,10] as well as exclusively non-quantitative metabolomics have been used to study predefined patient groups [11,12,13,14,15,16,17,18,19,20].

Recently, two studies reported a metabolomics approach for metabolic diagnostics in individual patients and argued their potential applicability in clinical diagnostics of IEM [1,21]. The workflow presented by Miller et al. is semi-quantitative and involves three separate mass spectrometry (MS) platforms run in parallel [21]. The workflow of Coene et al. is non-quantitative and involves a single-platform MS analysis [1]. Both workflows make use of chromatographic separation as they are based on either (high-resolution) gas chromatography MS (GC-MS) or (high-resolution) liquid chromatography MS (LC-MS) [1,21].

The optimal configuration for semi- and non-quantitative metabolomics in diagnostic applications is not known yet. In direct-infusion high-resolution MS (DI-HRMS), no chromatographic separation is performed, circumventing the need to create an experimental library containing metabolite masses and retention times and avoiding the problem of chromatographic alignment. For DI-HRMS, sample preparation is technically uncomplicated and only a very small amount of sample is needed (3 mm dried blood spot (DBS) or 20 µL plasma) [22]. Thus, DI-HRMS is in potential a powerful technique for the development of a high-throughput [23,24] broad test requiring limited patient material. DI-HRMS has been used in a quantitative and non-quantitative fashion for biomarker identification [25,26,27,28,29,30]. Moreover, Denes et al. started to explore its use for metabolic diagnostics by studying phenylketonuria, medium-chain acyl-coenzyme A dehydrogenase deficiency and homocystinuria [12]. However, in contrast to GC-MS and LC-MS approaches [1,21], the use of non-quantitative DI-HRMS has not been explored for a broad range of IEM.

In most metabolomics studies, the main focus is on plasma. However, DBS have the advantage of being easy to obtain (even at home), easy to store and easy to send and share. DBS have been used in some quantitative and non-quantitative chromatography based metabolomics approaches focusing on IEM [4,8,12,23]. Here, we describe a non-quantitative DI-HRMS metabolomics method for metabolic diagnostics. For the first time, we evaluate the use of a non-quantitative metabolomics method for both DBS and plasma by analyzing samples of patients with a broad range of IEM. As an example of the added clinical value, we present a case wherein DI-HRMS metabolomics aided interpretation of a variant of uncertain significance (VUS) identified by whole-exome sequencing (WES).

## 2. Results

### 2.1. Reproducibility Assessment

The variability within and between batches was analyzed to assess reproducibility of the acquired data. The samples of patients and controls were analyzed in seven batches by DI-HRMS (Figure 1). Between these batches there was limited variation in the number of mass peaks and metabolite annotations in the successive steps of data processing (Table 1). To monitor variability within each batch, a solution containing twenty stable isotope-labeled compounds (sILC) was added to each sample. Variability within each batch was satisfactory, with median relative standard deviations (RSD) of 0.20, 0.16, 0.16, 0.20, 0.21, 0.21, and 0.20, respectively (Table 2). To monitor variability between batches and to assess the variability in Z-scores, three DBS samples of three patients with IEM were analyzed in seven independent batches. Variability between batches was also satisfactory, with a median RSD of 0.24 for the seven compounds assessed. RSDs of the individual compounds were 0.19, 0.30, and 0.24 for propionylcarnitine, glycine and propionylglycine in the sample of the patient with propionic aciduria (PA), 0.15, 0.26 and 0.09 for citrulline, glutamine and lysine in the sample of the patient with lysinuric protein intolerance (LPI) and 0.33 for phenylalanine in the sample of the patient with phenylketonuria (PKU) (Table 3). Many other compounds annotated in these three patients, representing major chemical classes of metabolites such as acylcarnitines, amino acids, and organic acids, were found to have satisfactory RSDs (all data is available on request). Altogether, the reproducibility of the results pointed out by the limited variability within a batch and between different batches, indicates that all the components of the non-quantitative DI-HRMS metabolomics method, including resolution and sensitivity of the system, are stable over time.

### 2.2. Evaluation of the Clinical Value of the Method

For each patient sample, Z-scores were calculated for all mass peaks annotated with metabolites that can occur endogenously (~3835) (Figure 1). Based on these Z-scores and their ranking, a ‘most probable diagnosis’ was assigned to each patient (Table 4). The assigned diagnoses corresponded to the known diagnoses in 37/42 (88%) of patients based on DBS samples and in 32/38 (84%) of patients based on plasma samples. Figure 2 exemplifies Z-scores of biomarkers that had a very high Z-score and thus contributed to the most probable diagnoses (Figure 2). 

In the five patients in whom the assigned diagnosis was incorrect based on DBS samples, the biochemical profile had (almost) normalized due to treatment, as confirmed by routine targeted diagnostics in corresponding plasma samples of the patients. For example, a patient with 3-phosphoglycerate dehydrogenase (3-PGDH) deficiency presented with normalized serine due to serine supplementation (188, 234, and 83 μmol/mL, reference range 70–194 μmol/mL) and glycine (368, 213, and 161 μmol/mL, reference range 107–343 μmol/mL) in plasma. Serine Z-score in this patients’ DBS was 0.8, glycine Z-score −0.1 (Table 4). 

When not taking into account samples in which the biochemical profiles approximated the normal range due to treatment, most probable diagnoses were correct in all patients. Moreover, for all but one of the 23 included IEM we demonstrated that the IEM is correctly diagnosable in DBS using DI-HRMS.

In six patients the assigned diagnosis was incorrect based on plasma samples. In one patient the diagnosis 3-methylcrotonylglycinuria was missed, due to normal values of methylcrotonylglycine and hydroxyisovalerylcarnitine. In two patients with methylenetetrathydrofolate reductase (MTHFR) deficiency the intensity of the biomarker homocysteine was normal, not reflecting the concentration measured using targeted diagnostics. In a patient with non-ketotic hyperglycinaemia the biochemical profile had (almost) normalized due to treatment and lastly, in two patients carnitine palmitoyltransferase (CPT) I deficiency was missed since plasma might be a less adequate matrix than DBS for diagnosing these patients [31,32]. 

### 2.3. Direct-Infusion Based Metabolomics in Metabolic Diagnostics

As an example of the added clinical value of direct-infusion based metabolomics in metabolic diagnostics, we present a case wherein DI-HRMS metabolomics aided interpretation of a VUS identified by WES. A 16-year-old patient was referred by his psychiatrist to the genetics department because of a combination of an autism spectrum disorder, therapy resistant psychoses, intellectual disability (IQ 62-77), and some mild facial dysmorphisms. Routine targeted metabolic diagnostics revealed a severely decreased plasma l-carnitine (1.5 µmol/L, reference range 21.6–57.4 µmol/L). WES revealed a missense variant in the gene *TMLHE*, located on the X-chromosome (c.230G > A, p.R77Q, a hemizygous VUS). *TMLHE* encodes trimethyllysine hydroxylase (TMLHE), an enzyme that converts trimethyllysine into 3-hydroxytrimethyllysine. Mutations in *TMLHE* cause susceptibility to X-linked autism, type 6 (OMIM #300872). To support this possible diagnosis, we analyzed five DBS sampled at five different time points. The first two DBS were sampled prior to carnitine supplementation, DBS #3 and #4 during supplementation with 1000 mg levocarnitine and DBS #5 during supplementation with 1500 mg levocarnitine. At 3/5 time points trimethyllysine was significantly increased (Figure 3). Interestingly, at 5/5 time points, γ-butyrobetaine, a metabolite downstream of TMLHE and the precursor of l-carnitine, was significantly decreased (Figure 3). The substrate/(downstream-)product ratio trimethyllysine/γ-butyrobetaine was significantly increased in all five patient samples, clearly separating this patient from control samples and samples of other patients and thereby supporting pathogenicity of the identified VUS in the *TMLHE* gene (Figure 3). 

## 3. Discussion

We here present the first non-quantitative DI-HRMS metabolomics method for metabolic diagnostics and demonstrate its consistency and accuracy for both DBS and plasma. Using this method, 37/42 of the included patients could be correctly assigned a most probable diagnosis based on DBS samples, corresponding to 22/23 of included IEM. In addition, 32/38 included patients could be assigned a correct diagnosis based on plasma samples, corresponding to 19/21 of included IEM. For 3-PGDH we could not demonstrate the accuracy of the method in DBS, as the sample was drawn under serine supplementation, normalizing the level of serine. However, decreased concentrations of amino acids can be detected by this method, as illustrated by the repeatedly decreased concentration of lysine in a sample of a patient affected with LPI (Table 3 and Table 4). Our method failed to detect MTHFR deficiency and CPT1 deficiency in plasma. Methanol-based sample extraction as performed here only captures the unbound fraction of homocysteine, which is not a good representative of the total homocysteine concentration in blood, and therefore MTHFR deficiency was missed in plasma. Interestingly, in DBS MTHFR deficiency was correctly diagnosed since homocysteine thiolactone, an intramolecular thioester of homocysteine, was significantly increased (Table 2). CPT1 was missed in plasma, since we and others already demonstrated that DBS is a more suitable matrix to diagnose CPT1 deficiency [31,32], although diagnosis based on ratios may be possible in plasma [31,33].

In direct infusion, an observed mass can account for multiple metabolite annotations, since a specific *m*/*z* can account for multiple isomers (Figure 1). Here, this did not result in any data interpretation problems nor in any incorrect diagnosis, probably since most *m*/*z* are dominated by a single metabolite of significant abundance. Still, semi-quantitative metabolomics always produces tentative results. When used in daily practice, we advocate confirmation of the results with second-tier testing, using targeted diagnostic platforms or genetic tests. 

Direct infusion is superior to LC-MS methods in being nonselective and very sensitive and it allows the identification of many mass peaks. Even after strict metabolite selection, ~1.875 mass peaks corresponding to metabolites that can occur endogenously remained, compared to ~400 or ~340 clinically relevant metabolites in LC-MS and/or GC-MS based metabolomics methods [1,21]. The method of Miller et al. did not identify guanidinoacetic acid, methylmalonic acid and C14:1 and C14:2 carnitine [21]. In our method, these four metabolites were annotated, demonstrated by the correct diagnosis of guanidinoacetic acid methyltransferase deficiency, methylmalonic aciduria and very long-chain acyl-CoA dehydrogenase deficiency. The method of Coene et al. could not identify guanidinoacetic acid, argininosuccinic acid and dimethylglycine and thus could not recognize guanidinoacetic acid methyltransferase deficiency, argininosuccinic aciduria, and dimethylglycine dehydrogenase deficiency [1]. These last two IEM were not included in our study, but both argininosuccinic acid and dimethylglycine were annotated in every batch, suggesting that we should be able to correctly diagnose these IEM. Lastly, the method of Coene et al. could not recognize LPI due to strict criteria for the statistical tests [1]. We demonstrate that using our method we could diagnose LPI based on the combined decrease of lysine (mean Z-score −2.0, rank 7th from below) and increase of citrulline (mean Z-score 8.5, rank 2nd).

A strength of our study is that the assignment of the ‘most probable diagnosis’ was performed blindly, as the investigator was blinded to clinical data and to diagnoses included in the study. In addition, we demonstrate for the first time that next to plasma, as used in the studies of Miller and Coene et al. [1,21], DBS are also suitable material to use for non-quantitative metabolomics for metabolic diagnostics. Also, in contrast to Coene et al., who considered four peaks individually for each metabolite [1], we sum mass peak intensities of single adduct ions, leaving only one peak per metabolite to assess, facilitating data interpretation to a great extent. 

Potential clinical applications of non-quantitative metabolomics are numerous. We exemplified this by demonstrating a case in which DI-HRMS metabolomics aided interpretation of a VUS identified by WES. In addition, non-quantitative metabolomics can be used for generating hypotheses for new disease biomarkers, for studying pathophysiology in pre-defined patient groups and for studying the complete metabolome of cases suspected of an IEM that remained unresolved after routine metabolic diagnostics. 

## 4. Methods

### 4.1. Sample Collection

Blood samples were initially drawn for routine metabolic diagnostics or follow-up. Blood samples were collected by venous puncture in heparin-containing tubes. For DBS, aliquots of the blood sample were aspirated or a blood sample was collected by a finger prick. Blood samples were spotted onto Guthrie card filter paper (Whatman no. 903 Protein saver TM cards). The papers were left to dry for at least four hours at room temperature. DBS were stored at −80 °C in a foil bag with desiccant package pending further analysis. For plasma, blood samples were centrifuged at room temperature. The plasma was collected and stored at −80 °C pending further analysis. 

### 4.2. Patient Inclusion and Sample Selection

IEM of interest for this study were IEM with one or more known small-molecule biomarkers in blood, such as urea cycle disorders, organic acidurias, amino acid metabolism disorders, amino acid transport disorders, fatty acid oxidation disorders and disorders of creatine biosynthesis. Patients harboring such IEM were included when one or more residual DBS or plasma samples were available in the metabolic diagnostics laboratory of the University Medical Centre Utrecht, with a maximum of two patients per IEM. All patients or their legal guardians approved the possible use of their remnant samples for method development and validation, in agreement with institutional and national legislation. All procedures followed were in accordance with the ethical standards of the University Medical Centre Utrecht and with the Helsinki Declaration of 1975, as revised in 2000. 

Patient samples were included based on availability, and noteworthy, they were specifically neither selected based on whether a patient was under clinical management at time of sampling or not, nor based on presence of abnormal biochemical findings. 110 DBS from 42 patients with 23 different IEM and 86 plasma samples from 38 patients with 21 different IEM could be included (Figure 1). To serve as control samples, samples of individuals in whom an IEM was excluded after a thorough routine diagnostic work-up, were selected out of banked DBS and plasma samples. For DBS, 30 samples of 30 control individuals could be included. For plasma, 28 samples of 28 control individuals could be included (Figure 1). Control samples varied in age (including both neonates and elderly), sex, time of day at sampling, diets, supplements, drugs, storage time, and for control plasma samples degree of hemolysis, in order to take into account considerable variation in factors that might influence an individuals’ metabolome. Patient DBS samples were analyzed in four different batches, with each batch including all control DBS samples. Patient plasma samples were analyzed in three different batches, with each batch including all control plasma samples (Figure 1). In each batch, the samples were analyzed in randomized order. 

### 4.3. Sample Preparation

In order to be able to assess within batch variation, a working solution containing stable isotope-labeled compounds (sILC) was prepared in methanol achieving concentrations of 100 μM ^15^N; 2−^13^C-glycine, 20.0 μM ^2^H_4_-alanine, ^2^H_3_-leucine, ^2^H_3_-methionine, ^13^C_6_-phenylalanine, ^13^C_6_-tyrosine, ^2^H_3_-aspartate, ^2^H_3_-glutamate, ^2^H_2_-ornithine, ^2^H_2_-citrulline, and ^2^H_4_;^13^C-arginine, 6.08 μM ^2^H_9_-carnitine, 1.52 μM ^2^H_3_-acetylcarnitine, 0.304 μM ^2^H_3_-propionylcarnitine, ^2^H_3_-butyrylcarnitine, ^2^H_9_-isovalerylcarnitine, ^2^H_3_-octanoylcarnitine, ^2^H_9_-myristoylcarnitine, and 0.608 μM ^2^H_3_-palmitoylcarnitne (Cambridge Isotope Laboratories, Buchem b.v., Apeldoorn, The Netherlands). For DBS, this working solution was diluted 1:4. For plasma, sILC working solution was used undiluted. DBS sample extraction (3 mm, ~3.1 μL whole blood) was performed by addition of 140 μL sILC working solution, followed by a twenty minute ultra-sonication step. DBS samples were diluted with 60 μL 0.3% formic acid (Emsure, Darmstadt, Germany). Plasma samples were thawed to room temperature. Twenty μL of the plasma sample was added to 140 μL sILC working solution. This solution was centrifuged for five minutes at 17,000 g. One hundred and five microliters of supernatant was diluted with 45 μL 0.3% formic acid. Both DBS and plasma solutions were filtered using a methanol preconditioned 96 well filter plate (Acro prep, 0.2 μm GHP, NTRL, 1 mL well; Pall Corporation, Ann Arbor, MI, USA) and a vacuum manifold. The sample filtrate was collected in a 96 well plate (Advion, Ithaca, NY, USA).

### 4.4. DI-HRMS Analysis

A TriVersa NanoMate system (Advion, Ithaca, NY, USA) controlled by Chipsoft software (version 8.3.3, Advion) was mounted onto the interface of a Q-Exactive high-resolution mass spectrometer (Thermo Scientific™, Bremen, Germany). This system is a combined automatic sampler and nano-electrospray ionization (ESI) source. It houses a rack of 384 disposable conductive pipette tips and a 96-well microtiter plate that holds the samples to be analyzed. The ESI-Chip contains four hundred nozzles with nominal internal diameter of 5 μm. Samples (13 μL) were automatically aspirated sequentially into a pipette tip followed by an air gap (2 μL). For each sample, technical triplicates were analyzed, infusing each sample three times into the mass spectrometer (Figure 1). Pipette tips were engaged with the ESI-Chip to deliver the sample using nitrogen gas pressure at 0.5 psi and a spray voltage of 1.6 kV. For each sample new pipette tips and nozzles were used to prevent any cross-contamination or carryover. The Q-Exactive high-resolution mass spectrometer was operated in positive and negative ion mode in a single run, with automatic polarity switching. There were two time segments of 1.5 min with a total run time of 3.0 min. Scan range was 70 to 600 mass to charge ratio (*m*/*z*), resolution was 140,000 at *m*/*z* = 200 for optimal separation of *m*/*z*, automatic gain control target value was 3e6, maximum injection time was 200 ms, capillary temperature was 275 °C, the sample tray was kept at 18 °C and for the S-lens RF level factor 70 was used. In the positive segment, the sodium adduct of ^13^C_6_-Phenylalanine (*m*/*z* 194.0833) was used as a lock mass and in the negative segment ^2^H_9_-myristoylcarnitine (*m*/*z* 381.36733) was used as lock mass. 

### 4.5. Data Processing

Data acquisition was performed using Xcalibur software (version 3.0, Thermo Scientific™, Waltham, MA, USA). Using MSConvert15 (ProteoWizard Software Foundation, Palo Alto, CA, USA), raw data files containing scanning time, *m*/*z* and peak intensity were converted to mzXML format in Profile mode. A peak calling pipeline was developed in R programming language. Peak calling of raw mass spectrometry data resulted in ~186,000 mass peaks per batch (Figure 1), with for each mass peak the mean intensity of the technical triplicate (Figure 1), excluding miss infusions. Detected mass peaks were annotated by matching the *m*/*z* of the mass peak with a range of two parts per million to metabolite masses present in the Human Metabolome Database (HMDB), version 3.6 [34]. Taking into account isomers and adduct ions, ~60,000 mass peaks could be annotated with one or more possible identifications (Figure 1). 

Since during electrospray ionization adduct ions are formed, each of the ~60,000 annotated mass peaks may account for a metabolite without adduct ion, or with one or more adduct ions. This means that one metabolite can be detected with different adduct ions, thus different mass peaks. To facilitate interpretation, metabolite annotations without adduct ions in negative or positive mode ([M − H]^–^, [M + H]^+^), or with the single adduct ions [M + Na]^+^, [M + K]^+,^ and [M + Cl]^−^ were selected. For each sample, the intensities of these five mass peaks were summed, resulting in one (summed) mass peak intensity per metabolite annotation: ~6600 summed mass peaks in total. Next, exogenous and drug metabolite annotations were excluded by manual curation of the database by four trained clinical laboratory geneticists. Metabolites that can occur endogenously and metabolites with still unknown function were included, resulting in ~1875 summed mass peaks in total, corresponding to ~3835 metabolite annotations (Figure 1), since mass peaks can account for several isomers. Data and R code can be supplied upon request.

### 4.6. Data Analysis

Data processing provided the mean intensity of the technical triplicate per mass peak, per sample (Figure 1). Since non-quantitative metabolomics provides relative comparisons between samples, interpretation of the mass peak intensities was enabled by calculating the mean intensity and the standard deviation of the intensities of the control samples in the same batch for each mass peak (Figure 1). For each mass peak per patient sample, the deviation from the intensities in control samples was indicated by a Z-score, calculated by: Z-score = (intensity patient sample—mean intensity control samples)/standard deviation intensity control samples (Figure 1). For each sample, metabolite annotations were ranked on Z-score: positive Z-scores were ranked from rank 1 onwards from the highest Z-scores to a Z-score of 0.0, and negative Z-scores were ranked from rank (−)1 onwards from the lowest Z-scores to a Z-score of 0.0. 

### 4.7. Evaluation of the Clinical Value of the Method

For each patient, mean intensity and mean Z-score were calculated over the included patient samples. Each patient was assigned a ‘most probable diagnosis’ based on metabolite Z-score and rank per patient sample and mean metabolite Z-score and rank. The investigator assigning these diagnoses was blinded for the patients’ true diagnosis, and for the list of 23 included IEM. Furthermore, neither information regarding patient characteristics (sex, age), nor clinical information was provided to the researcher. 

### 4.8. Reproducibility Assessment

To monitor variability within each batch, twenty sILC were added to each sample. For each sILC, the relative standard deviation (RSD) was calculated over all samples in a batch, calculated by: standard deviation intensity/mean intensity. A median RSD below 0.25 was considered satisfactory. To monitor variability between batches and to assess the variability in Z-scores, three DBS samples of three patients, one with PA, one with LPI and one with PKU were analyzed in seven batches. For PA, metabolites of interest were propionylcarnitine, glycine, and propionylglycine. For LPI, metabolites of interest were citrulline, glutamine, and lysine and for PKU the metabolite of interest was phenylalanine. For each of these seven metabolites, the RSD was calculated over all seven samples measurements, calculated by: standard deviation Z-score/mean Z-score. A median RSD below 0.25 was considered satisfactory. 

## 5. Conclusions

In summary, we demonstrate a very consistent, nonselective, high-throughput DI-HRMS metabolomics method for investigating the metabolome in genetic disease, which requires only small amounts of patient material. We demonstrated the value of this method for both DBS and plasma by blindly assigning a correct ‘most probable diagnosis’ to patients with a wide range of IEM. 

## Figures and Tables

**Figure 1 metabolites-09-00012-f001:**
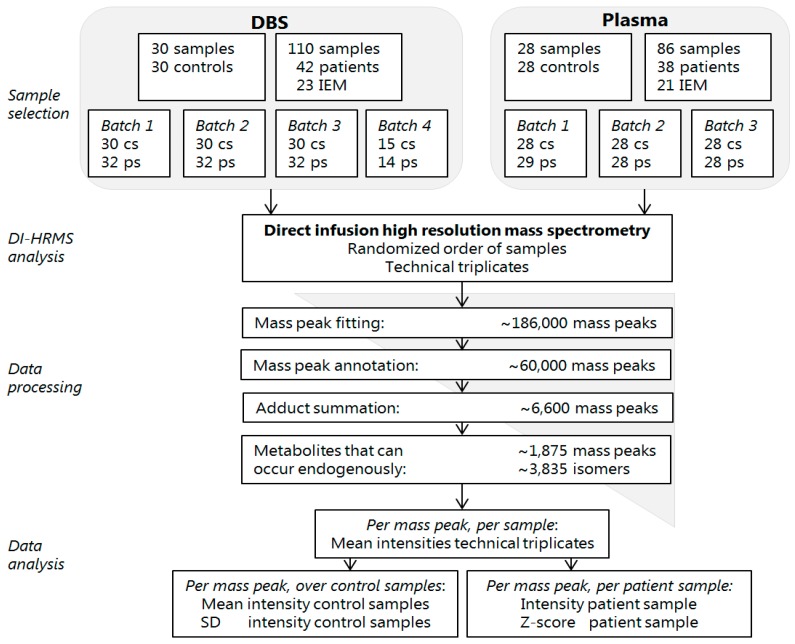
Flowchart of the non-quantitative direct-infusion high-resolution MS (DI-HRMS) method. DBS: dried blood spots, IEM: inborn error of metabolism, cs: control samples, ps: patient samples, DI-HRMS: direct infusion high resolution mass spectrometry, SD: standard deviation.

**Figure 2 metabolites-09-00012-f002:**
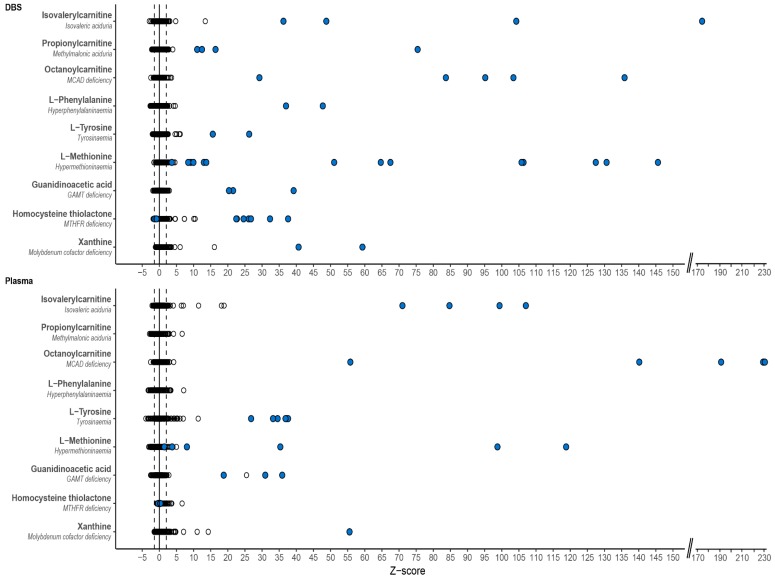
Representative examples of important biomarkers for the diagnosed IEM. For both dried blood spots (DBS) and plasma three acylcarnitine biomarkers, three amino acids and three other metabolites are demonstrated as representative examples of compounds with high Z-scores that contributed to the assigned most probable diagnosis. Each dot represents a unique sample. Black open circles represent both control samples and samples of other IEM patients, blue filled circles represent patients with that specific IEM. MCAD: medium-chain acyl-CoA dehydrogenase, GAMT: guanidinoacetate methyltransferase; MTHFR: methylenetetrahydrofolate reductase.

**Figure 3 metabolites-09-00012-f003:**
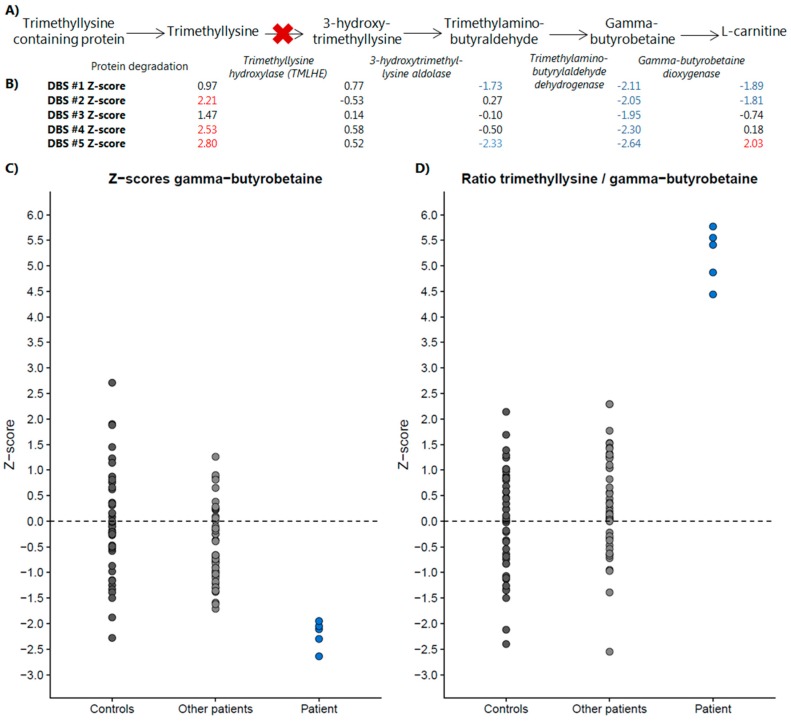
DI-HRMS metabolomics pathway analysis of the pathway of trimethyllysine hydroxylase. (**A**) Metabolic pathway of trimethyllysine hydroxylase, in italic the enzyme responsible for the metabolite conversion depicted by the arrow. (**B**) Z-scores of the five included dried blood spots (DBS) for each of the metabolites included in the pathway. Z-scores in red are significantly increased, Z-scores in blue are significantly decreased. (**C**) Z-scores of control samples, other patient samples and the samples of this patient for γ-butyrobetaine, the metabolite decreased in all patient samples. (**D**) Z-scores of control samples, other patient samples and the samples of this patient for the ratio of trimethyllysine over γ-butyrobetaine. De ratio is significantly increased in all patient samples and clearly separates this patient from control samples and samples of other patients, thereby supporting pathogenicity of the identified variant of uncertain significance (VUS) in the *TMLHE* gene.

**Table 1 metabolites-09-00012-t001:** Variation in number of mass peaks and metabolite annotations in data processing steps.

	DBS				Plasma		
	Batch 1	Batch 2	Batch 3	Batch 4	Batch 1	Batch 2	Batch 3
Mass peak fitting	185,661	176,934	197,681	190,172	192,198	177,879	185,642
Mass peak annotation	59,543	56,250	63,360	60,979	62,503	58,212	60,450
Adduct summation	6580	6625	6598	6611			
Endogenous mass peaks *	1874	1885	1874	1875	1875	1867	1868
Endogenous metabolite annotations *	3822	3863	3826	3839	3832	3847	3817

* Mass peaks and annotations corresponding to metabolites that can occur endogenously.

**Table 2 metabolites-09-00012-t002:** Within-batch variation: relative standard deviation of stable isotope-labeled compounds.

	DBS				Plasma		
	*Batch 1*	*Batch 2*	*Batch 3*	*Batch 4*	*Batch 1*	*Batch 2*	*Batch 3*
^15^N;2−^13^C-glycine	0.23	0.16	0.18	0.24	0.22	0.21	0.79
^2^H_4_-alanine	0.20	0.14	0.16	0.20	0.20	0.21	0.19
^2^H_3_-leucine	0.18	0.14	0.15	0.18	0.60	0.55	0.50
^2^H_3_-methionine	0.31	0.30	0.36	0.39	1.70	0.22	0.20
^13^C_6_-phenylalanine	0.19	0.16	0.14	0.18	0.21	0.20	0.19
^13^C_6_-tyrosine	0.19	0.17	0.16	0.20	0.22	0.21	0.18
^2^H_3_-aspartate	0.24	0.22	0.22	0.25	0.23	0.24	0.26
^2^H_3_-glutamate	0.17	0.15	0.14	0.18	0.20	0.21	0.15
^2^H_2_-ornithine	0.21	0.19	0.17	0.21	0.14	0.17	0.12
^2^H_2_-citrulline	0.16	0.16	0.14	0.18	0.18	0.19	0.14
^2^H_4_;^13^C-arginine	0.21	0.17	0.16	0.20	0.17	0.18	0.16
^2^H_8_-valine	0.18	0.14	0.15	0.18	0.20	0.19	0.18
^2^H_9_-carnitine	0.27	0.21	0.22	0.30	0.22	0.24	0.21
^2^H_3_-acetylcarnitine	0.89	0.21	0.82	0.92	0.46	0.46	0.74
^2^H_3_-propionylcarnitine	0.21	0.16	0.16	0.20	0.19	0.20	0.20
^2^H_3_-butyrylcarnitine	3.39	0.63	1.34	1.53	0.77	0.92	1.08
^2^H_9_-isovalerylcarnitine	0.20	0.13	0.15	0.17	0.19	0.20	0.19
^2^H_3_-octanoylcarnitine	0.18	0.12	0.14	0.17	0.16	0.21	0.20
^2^H_9_-myristoylcarnitine	0.20	0.14	0.14	0.17	0.20	0.22	0.20
^2^H_3_-palmitoylcarnitne	0.19	0.16	0.15	0.18	0.23	0.23	0.21
5th percentile	0.16	0.13	0.14	0.17	0.16	0.18	0.14
Median	0.20	0.16	0.16	0.20	0.21	0.21	0.20
95th percentile	2.96	0.30	0.79	0.89	0.76	0.55	0.79

**Table 3 metabolites-09-00012-t003:** Between-batch variation: relative standard deviation of Z-scores of positive control samples.

	*Batch 1*	*Batch 2*	*Batch 3*	*Batch 4*	*Batch 5*	*Batch 6*	*Batch 7*	RSD
*Propionic aciduria*								
Propionylcarnitine	40.23	66.57	47.17	70.07	61.18	52.29	52.66	0.19
Glycine	12.99	20.75	17.28	16.42	23.48	24.54	10.12	0.30
Propionylglycine	7.89	7.69	9.26	7.54	12.69	9.09	6.10	0.24
*Lysinuric protein intolerance*								
Citrulline	23.86	26.55	18.98	29.23	24.10	30.03	22.51	0.15
Glutamine	3.32	3.40	3.56	4.68	3.39	4.81	2.07	0.26
Lysine	−2.07	−2.13	−1.89	−2.07	−2.25	−1.97	−1.71	0.09
*Phenylketonuria*								
Phenylalanine	34.29	17.93	16.79	23.13	21.74	16.21	14.19	0.33

**Table 4 metabolites-09-00012-t004:** Assigned most probable diagnoses based on metabolite Z-scores in DBS and plasma.

			DBS #1			DBS #2			Plasma #1			Plasma #2		
	Patient Diagnosis	Metabolite *	Z-sc.	Rank	Correct Diagn.	Z-sc.	Rank	Correct Diagn.	Z-sc.	Rank	Correct Diagn.	Z-sc.	Rank	Correct Diagn.
Urea cycle	OTC deficiency	Orotic acid	5.7	1	Yes (*n* = 2)	−0.5		No (*n* = 3)	11.7	2	Yes (*n* = 2)	2.3		Yes (*n* = 1)
		Uridine	1.6			−0.5			7.1			39.2	6	
		5-Oxoproline	−0.7			0.2			0.4			9.0		
		Uracil	−0.8			−0.7			4.0			4.0		
		Orotidine	0.1			−0.5			−1.1			4.0		
		L-Lysine	0.0			−0.2			0.3			3.3		
		Citrulline	−0.3			−1.8	−7		−0.6			−2.8	−7	
Branched-chain amino acid metabolism	MSUD	Ketoleucine	23.3	2	Yes (*n* = 4)	3.0	20	Yes (*n* = 3)	65.0	7	Yes (*n* = 4)	13.3	17	Yes (*n* = 3)
		(*n* = Iso) leucine	12.4	6		0.4			37.4	10		24.7	7	
		2-Hydroxy-3-methylbutyr. acid	9.4	8		−0.4			579.1	1		234.8	1	
		Alpha-ketoisovaleric acid	4.8			2.2			39.5	9		21.2	9	
	IVA	Isovalerylcarnitine	137.9	1	Yes (*n* = 2)	42.5	2	Yes (*n* = 2)	84.7	1	Yes (*n* = 1)	92.5	1	Yes (*n* = 3)
		3-Hydroxyisovaleric acid	0.0			−0.1			0.4			0.1		
	3-MCC	3-Hydroxyisovaleric acid	5.4	1	Yes (*n* = 2)	33.1	1	Yes (*n* = 2)	17.8	4	No (*n* = 1)	825.8	1	Yes (*n* = 2)
		3-Methylcrotonylglycine	0.7			22.8	2		−0.1			2.7		
		Isovalerylcarnitine	0.6			−1.2			7.0	11		0.7		
		3-Hydroxyisovalerylcarnitine	0.6			−1.6			−0.2			0.0		
	MMA	Propionylcarnitine	13.3	2	Yes (*n* = 3)	75.4	1	Yes (*n* = 1)						
		Methylcitric acid	7.3	4		4.3								
		Methylmalonic acid	0.2			16.6	4							
		Methylmalonylcarnitine	1.1			0.7								
Lysine metabolism	GA-1	Glutarylcarnitine	18.6	2	Yes (*n* = 4)	4.9	10	Yes (*n* = 2)	26.3	1	Yes (*n* = 3)	27.9	5	Yes (*n* = 3)
		Glutaric acid	7.9	3		−0.9			6.4	17		71.6	3	
		3-hydroxyglutaric acid	−0.3			0.2			10.5	8		8.2	11	
		Glutaconic acid	−1.64			27.9	2							
Phenylalanine and tyrosine metabolism	PKU	Phenylalanine	47.7	1	Yes (*n* = 1)	37.0	3	Yes (*n* = 1)						
		Hydroxyphenylacetic acid	10.9	4		1.9								
		N-acetylphenylalanine	6.3	9		7.0	22							
		Tyrosine	−1.0			−0.1								
	Tyrosinaemia	4-Hydroxyphenyllactic acid	150.7	1	Yes (*n* = 1)	125.6	2	Yes (*n* = 1)	206.5	1	Yes (*n* = 3)	263.5	13	Yes (*n* = 3)
		Tyrosine	26.2	3		15.6	6		35.0	3		33.7		
		4-Hydroxyphenylacetic acid	4.6			6.3	9		2.2			2.0		
		4-Hydroxyphenylpyruvic acid	0.2			2.0			10.4	8		6.8		
		Succinylacetone	−1.5			−1.2			0.2			1.1		
Sulphur amino acid metabolism	MAT1A deficiency	Methionine sulfoxide	72.2	1	Yes (*n* = 5)	53.4	2	Yes (*n* = 5)	1106.7	1	Yes (*n* = 1)	632.2	1	Yes (*n* = 3)
		Methionine	57.1	2		96.4	1		118.8	4		47.4	6	
		S-adenosylmethionine	0.5			−0.3			0.1			0.3		
		S-adenosylhomocysteine	−0.8			0.4			0.4			0.1		
	CBS deficiency	Methionine sulfoxide	22.4	2	Yes (*n* = 4)				778.9	1	Yes (*n* = 2)			
		Methionine	31.1	3					2.6					
		Homocystine	3.2	7					1.3					
		Homocysteine	2.6	12					2.2					
	MTHFR deficiency	Homocysteine thiolactone	28.0	1	Yes (*n* = 6)	7.5	3	Yes	−0.3		No (*n* = 1)	0.1		No (*n* = 3)
		Homocystine	1.1			4.8	9	(*n* = 3)	−0.2			0.3		
		Methionine	0.2			0.0			−2.4	−20		−2.3	−12	
	Molybdenum cofactor deficiency	Xanthine	59.3	1	Yes (*n* = 1)	40.7	3	Yes	55.5	7	Yes (*n* = 1)			
		Alpha amino adipic semialdeh.	3.4			1.5		(*n* = 1)	6.9					
		Cysteine-S-sulfate	−0.9			0.6			11.8	22				
		Cysteine	−1.0			−2.6			−2.1	−14				
		Uric acid	−1.4			−0.8			−2.6	−5				
Serine and glycine metabolism	NKH	Glycine	3.7	18	Yes (*n* = 2)	2.0		No (*n* = 3)	3.4		Yes (*n* = 3)	2.2		No (*n* = 3)
	3-PGDH deficiency	Serine	5.1	1	No (*n* = 3)	0.8		No	−2.5	−4	Yes (*n* = 2)	−2.4	−6	Yes (*n* = 2)
		Glycine	2.1			−0.1		(*n* = 3)	−1.6			−1.8		
Proline metabolism	OAT deficiency	Proline	4.0	11	Yes (*n* = 6)				4.0		Yes (*n* = 5)			
		Ornithine	2.8	18					−0.8					
Amino acid transport	LPI	Citrulline	8.5	2	Yes (*n* = 3)				16.1	13	Yes (*n* = 3)			
		Serine	6.2	3					2.4					
		Proline	6.4	4					0.2					
		Threonine	5.7	7					0.6					
		Lysine	−2.0	−7					−1.3					
		Ornithine	−1.5						0.6					
		Arginine	−1.0						−1.0					
Fatty acid oxidation	VLCAD deficiency	C14:1 carnitine	28.9	1	Yes (*n* = 1)	0.6		No (*n* = 3)	7.3	34	Yes (*n* = 1)	5.8		Yes (*n* = 1)
		C14:2 carnitine	15.7	2		1.4			7.6	33		2.8		
		C14-carnitine	3.7			1.5			1.4			2.4		
	LCHAD deficiency	C14-OH carnitine	3.1	35	Yes (*n* = 1)				8.3	14	Yes (*n* = 2)	8.2		Yes (*n* = 2)
		C16-OH carnitine	3.0	37					22.7	2		37.3	12	
		C16-OH:1 carnitine	1.5						23.8	1		41.6	11	
		C18-OH carnitine	0.7						21.9	3		29.8	17	
	MCAD deficiency	C8-carnitine	56.5	1	Yes (*n* = 2)	111.5	1	Yes (*n* = 3)	189.3	1	Yes (*n* = 3)	143.4	1	Yes (*n* = 2)
		C6-carnitine	7.3	6		16.0	3		51.7	2		55.7	2	
		C10:1-carnitine	1.7			8.1	7		24.9	4		11.6	5	
		C10-carnitine	1.1			2.6			7.3	12		3.2		
	OCTN2 deficiency	L-Carnitine	−2.0		Yes (*n* = 1)	−1.3		Yes (*n* = 4)	−2.4	−3	Yes (*n* = 2)	−2.3	−6	Yes (*n* = 1)
		Acetylcarnitine	−1.9			−0.9			−2.5	−1		−2.5	−9	
		C16-carnitine	−1.7			−1.3			−1.1			−0.3		
		C16:1-carnitineC18-carnitine	−2.6–1.7	−5		−1.1–1.7	−2		−1.3–0.6			−1.8–0.9		
		C18:1-carnitine	−2.3	−12		−1.8	−1		−1.1			−1.0		
	CPT1 deficiency	L-Carnitine	19.0	1	Yes (*n* = 2)	19.0	1	Yes (*n* = 6)	−2.7	−84	No (*n* = 2)	1.8		No (*n* = 2)
		C0/(*n* = C16 + C18) ratio	10.3	3		8.4	3		−1.6			−0.3		
		C16-carnitine	−3.1	−1		−1.8	−5		−2.7	−82		−0.2		
		C18-carnitine	−2.6	−3		−2.2	−2		−1.1			−0.6		
		C18:1-carnitine	−2.6	−4		−2.5	−1		0.0			1.1		
	CPT2 deficiency	C16+C18:1/C2 ratio	2.2	25	Yes (*n* = 2)	4.8	1	Yes (*n* = 3)	−1.4		Yes (*n* = 4)	0.1		Yes (*n* = 2)
		Acetylcarnitine	−1.7	−8		−2.4	−1		8.8	9		6.5	6	
		C16-carnitine	−0.6			−1.4			9.3	8		6.7	5	
		C18-carnitine	−0.6			−1.7			4.1			3.1		
		C18:1-carnitine	−0.7			−1.8								
Creatine biosynthesis	GAMT deficiency	Guanidoacetic acid	20.9	1	Yes (*n* = 2)	39.2	2	Yes (*n* = 1)	25.1	1	Yes (*n* = 3)	35.9	1	Yes (*n* = 1)
		Creatine	−1.4			−1.2			1.8			−1.7		

Metabolites that contributed most to assigning the most probable diagnosis are reported, although more metabolites have influenced the final decision. *For each mass peak, only one metabolite annotation is reported, the one that influenced the final decision. Metabolites that are not reported were either less relevant for assigning most probable diagnosis or were normal. All data is available on request. Z-sc. is the average Z-score of the included samples. The rank is the metabolite rank in the list of metabolites ordered on highest to lowest Z-score. A negative rank is the rank from bottom to top of the list. OTC: ornithine transcarbamylase; def.: deficiency; MMA: methylmalonic acidemia, isolated; GA−1: glutaric acidemia type I; IVA: isovaleric acidemia; 3-MCC: 3-methylcrotonylglycinuria; MSUD: maple syrup urine disease; PKU: phenylketonuria; MAT1A: methionine adenosyltransferase Ia; CBS: cystathionine beta-synthase; MTHFR: methylenetetrahydrofolate reductase; NKH: non-ketotic hyperglycinaemia; 3-PGDH: 3-phosphoglycerate dehydrogenase; OAT: ornithine aminotransferase; LPI: lysinuric protein intolerance; VLCAD: very long chain acyl-CoA dehydrogenase; LCHAD: long-chain hydroxyacyl-CoA dehydrogenase; MCAD: medium-chain acyl-CoA dehydrogenase; OCTN2: organic cation transporter 2; CPT1: carnitine palmitoyltransferase I; CPT2: carnitine palmitoyltransferase II; GAMT: guanidinoacetate methyltransferase.

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
