# Peer review of "Direct Infusion Based Metabolomics Identifies Metabolic Disease in Patients’ Dried Blood Spots and Plasma"

_metabolites, 2019, doi:10.3390/metabo9010012_

Round 1

Reviewer 1 Report

Authors submitted manuscript describing progressive and promising method that may be further elaborated in to the potential clinical application

The title of submitted manuscript - Direct infusion based metabolomics identifies metabolic disease in patients’ dried blood spots and plasma - directs readers at first to examine recent achievements in metabolomics and dried blood spots. DI-HRMS method is described and referenced. DBS utilization in newborn screening is well known in medical practice and referenced in this manuscript. However, the introduction does not provide sufficient background and does not include all relevant references. Simple search in Google with "DBS and metabolomics" can direct to very recent publications in per reviewed scientific journals.

Page 1

Abstract

....A peak calling pipeline developed in R programming language provided Z-scores for ~1,875 mass peaks 25 corresponding to ~3,835 metabolite identifications (including isomers) per sample.....

Word identifications should be replaced with annotations. It is the correct name for described pipeline protocol. 

Chapter Fifteen - Metabolite Annotation and Identification                                                         JoannaGodzien, AlbertoGil de la Fuente, AbrahamOtero, et al. https://doi.org/10.1016/bs.coac.2018.07.004

Assignation of signals covers two distinct processes: annotation and identification. Metabolite annotation is the assignment of tentative metabolite candidates to the signals based on matching their masses with database… Meanwhile, proper identification always entails the comparison with an authentic standard….

Metabolite identification is obviously critical step for clinical assay development.

Below are an examples of annotation but not identification available with DI-HRMS:

Leucine, isoleucine, alanine betaine, betaleucine, alloisoleucine, aminocaproic acid:  C4H9N3O2      

These molecules have the same elemental composition and accurate mass.

N-acetylalanine, propionyl glycine, hydroxyproline, aminolevulinic acid, oxoaminopentanoic acid, N-acetyl β-alanine, glutamate semi-aldehyde:   C5H9NO3

These molecules have the same elemental composition and accurate mass.

Hexoses - glucose, galactose and fructose are indistinguishable with DI-HRMS: C6H12O6 

Page 5

Figure 1, flowchart...

Replace word Endogenous with word Detected or Substances or Annotated, etc.. since these mass peaks are not all endogenous, as it is mentioned on page 10 reporting on patients amino acids supplementation. There is no list of drugs, supplements and diets of the study participants presented in the manuscript. Moreover, it is well known that direct infusion of plasma/blood extracts into the mass spectrometer does not discriminate isomers either positioned or chiral, and detected annotated molecules are mostly not individual metabolites but rather unresolved mixture of isomers. No MS/MS experiments were conducted for annotation as well.

Page 13

Please provide correct information: An approval from an ethics committee should have been obtained before undertaking the research with human subjects, human material, human tissues, or human data. At a minimum, a statement including the project identification codedate of approval and name of the ethics committee or institutional review board should be cited in the Methods Section of the article.

Page 14 

Mass peak identification and annotation was conducted by matching the m/z of the mass peak

with a range of two parts per million to metabolite masses present in the Human Metabolome Database (HMDB),

This sentence should be rephrased since it is misleading. Suggested rephrasing:

Detected mass peaks were annotated by matching the m/z of the ...... 

None of the "identified" metabolites in this study confirmed with the spiking experiment using an authentic standards. No MS/MS experiments were conducted for annotation as well.

Page 15

Metabolite identifications were ranked on Z-score. 

Please clarify this claim.  

Page 15

Conclusions:    ......blindly diagnosing patients with a wide range of IEM....

This is incorrect. Patient can be diagnosed with an approved and validated assay. Authors discriminated and characterized samples taken from study participants and described method for investigating the metabolome in genetic diseases, but not diagnosed patients. 

Author Response

Reviewer 1

Authors submitted manuscript describing progressive and promising method that may be further elaborated in to the potential clinical application. The title of submitted manuscript - Direct infusion based metabolomics identifies metabolic disease in patients’ dried blood spots and plasma - directs readers at first to examine recent achievements in metabolomics and dried blood spots. DI-HRMS method is described and referenced.

Point 1. DBS utilization in newborn screening is well known in medical practice and referenced in this manuscript. However, the introduction does not provide sufficient background and does not include all relevant references. Simple search in Google with "DBS and metabolomics" can direct to very recent publications in peer reviewed scientific journals.

Response to point 1. We thank the reviewer for this valid notion. We performed a new search into the use of DBS for metabolomics approaches focusing on IEM and added the study of Jacob et al. 2018 who indeed also describe a metabolomics method using DBS for the diagnosis of IEM. We now describe the use of DBS in these metabolomics approaches more explicitly in the last paragraph of the introduction: “DBS have been used in some quantitative and non-quantitative chromatography based metabolomics approaches focusing on IEM [4,8,12,23].” We do not explicitly reference all metabolomics studies using DBS that do not focus on IEM, since we feel the risk of being incomplete is too high with so many studies to refer to.

Point 2. Page 1, Abstract. “A peak calling pipeline developed in R programming language provided Z-scores for ~1,875 mass peaks 25 corresponding to ~3,835 metabolite identifications (including isomers) per sample”. Word identifications should be replaced with annotations. It is the correct name for described pipeline protocol. Assignation of signals covers two distinct processes: annotation and identification. Metabolite annotation is the assignment of tentative metabolite candidates to the signals based on matching their masses with database. Meanwhile, proper identification always entails the comparison with an authentic standard. Metabolite identification is obviously critical step for clinical assay development. Below are an examples of annotation but not identification available with DI-HRMS:

Leucine, isoleucine, alanine betaine, betaleucine, alloisoleucine, aminocaproic acid:  C4H9N3O2. These molecules have the same elemental composition and accurate mass. N-acetylalanine, propionyl glycine, hydroxyproline, aminolevulinic acid, oxoaminopentanoic acid, N-acetyl β-alanine, glutamate semi-aldehyde: C5H9NO3. These molecules have the same elemental composition and accurate mass. Hexoses - glucose, galactose and fructose are indistinguishable with DI-HRMS: C6H12O6.

Response to point 2. We thank the reviewer for this extensive explanation and we agree that the word annotation is more correct than the word identification. We replaced identification for annotation throughout the manuscript and we feel it is a significant improvement to the manuscript.

Point 3. Page 5. Figure 1, flowchart. Replace word Endogenous with word Detected or Substances or Annotated, etc.. since these mass peaks are not all endogenous, as it is mentioned on page 10 reporting on patients amino acids supplementation. There is no list of drugs, supplements and diets of the study participants presented in the manuscript. Moreover, it is well known that direct infusion of plasma/blood extracts into the mass spectrometer does not discriminate isomers either positioned or chiral, and detected annotated molecules are mostly not individual metabolites but rather unresolved mixture of isomers. No MS/MS experiments were conducted for annotation as well.

Response to point 3. We agree that the metabolites that we selected as “endogenous” are not by definition all endogenous metabolites, so we replaced “endogenous” throughout the manuscript for “metabolites that can occur endogenously”, including in Figure 1.

Point 4. Page 13. Please provide correct information: An approval from an ethics committee should have been obtained before undertaking the research with human subjects, human material, human tissues, or human data. At a minimum, a statement including the project identification code, date of approval and name of the ethics committee or institutional review board should be cited in the Methods Section of the article.

Response to point 4. According to institutional (University Medical Centre Utrecht) and national (the Netherlands) legislation, patients approve or disapprove upon sampling whether remnant samples can be used for method development and validation. All included patients, or their legal guardians, approved the possible use of their remnant samples. Since no further approval from an ethics committee is needed for studies using remnant samples in the Netherlands, there is no project identification code or date of approval from the ethics committee.

Point 5. Page 14.Mass peak identification and annotation was conducted by matching the m/z of the mass peak with a range of two parts per million to metabolite masses present in the Human Metabolome Database (HMDB). This sentence should be rephrased since it is misleading. Suggested rephrasing:

Detected mass peaks were annotated by matching the m/z of the… None of the "identified" metabolites in this study confirmed with the spiking experiment using an authentic standards. No MS/MS experiments were conducted for annotation as well.

Response to point 5. In line with point 2, we agree that the word annotation is more correct than the word identification. We rephrased the sentence as suggested.

Point 6. Page 15. Metabolite identifications were ranked on Z-score. Please clarify this claim.  

Response to point 6. We clarified this claim in the text as follows: “For each sample, metabolite annotations were ranked on Z-score: positive Z-scores were ranked from rank 1 onwards from the highest Z-scores to a Z-score of 0, and negative Z-scores were ranked from rank (-)1 onwards from the lowest Z-scores to a Z-score of 0.”

Examples can be found in Table 4, where the highest Z-score (5.7) for orotic acid in OTC deficiency ranks first, and a low Z-score (-1.8)  for citrulline ranks (-)7, indicating that in this sample there were six mass peaks with an metabolite annotation that could occur endogenously with lower Z-scores than citrulline.

Point 7. Page 15. Conclusions: ...blindly diagnosing patients with a wide range of IEM.... This is incorrect. Patient can be diagnosed with an approved and validated assay. Authors discriminated and characterized samples taken from study participants and described method for investigating the metabolome in genetic diseases, but not diagnosed patients.

Response to point 7. We agree that a final diagnosis for a patient can only be made using an approved and validated assay and that we demonstrate that we can characterize/discriminate samples of patients with known IEM. We rephrased ‘diagnosis’ throughout the manuscript to ‘most probable diagnosis’ and we rephrased the conclusion as follows: ”We demonstrated the value of this method for both DBS and plasma by blindly assigning a correct ‘most probable diagnosis’ to patients with a wide range of IEM.”

Reviewer 2 Report

In this study, the authors developed a direct infusion non-quantitative metabolomic method on a Q-Exactive high-resolution mass spectrometer. They demonstrated the utility of this method for the diagnosis of inborn errors of metabolismIEM).  The following suggestions are offered to help improve the manuscript:

(1)  The untargeted metabolomic analysis in this study led to the identification of 1875 endogenous compounds and 3835 isomers. However, only the selected known biomarkers were used for the diagnosis of IEM (Figure 2 and Table 4). How about a broad-spectrum targeted analysis of the known biomarkers for metabolic diseases?  A multivariate method such as PLS-DA, Random Forest, or PCA may yield some interesting and useful results.  These are conveniently implemented using the statistical packages available at: https://www.metaboanalyst.ca.

(2)  The reproducibility of the method was assessed using 20 isotope standards. Twenty compounds are too few to represent the total 1875 identified compounds and 3835 isomers in the samples. Most of the 20 isotopes are amino acids and acylcarnitines. These compounds are usually abundant in plasma and DBS. How about the RSD of compounds with low abundance in the samples?  For example, orotic acid, a key biomarker for OTC deficiency?  How about methylmalonic acid for Vitamin B12 deficiency or methylmalonic acidemia? More diverse representation of the major chemical classes should be selected as isotope standards to evaluate the reproducibility of the method in future implementations of these methods.

(3)  In Table 4, the authors showed the z-score of methylmalonic acid (MMA).  How did you accurately identify MMA? How do you avoid the interference of the atomically identical (C4H6O4) and structurally related isomer succinic acid? Both compounds have a monoisotopic mass of 118.026611 Daltons. Succinic acid is usually present in samples at a greater concentration than MMA.

(4)  Subjects metadata (age, sex, etc) is missing for this study. These metadata are essential for other researchers to use this method and compare the results across the studies.

(5)  In the method section (4.1), DBS were prepared either by spotting the blood taken from heparin-containing tubes or by the finger prick. The concentration of certain metabolites is different between these 2 approaches. In the future, a more standardized collection protocol should be used.

Author Response

Reviewer 2

In this study, the authors developed a direct infusion non-quantitative metabolomic method on a Q-Exactive high-resolution mass spectrometer. They demonstrated the utility of this method for the diagnosis of inborn errors of metabolism (IEM).  The following suggestions are offered to help improve the manuscript.

Point 1. The untargeted metabolomic analysis in this study led to the identification of 1875 endogenous compounds and 3835 isomers. However, only the selected known biomarkers were used for the diagnosis of IEM (Figure 2 and Table 4). How about a broad-spectrum targeted analysis of the known biomarkers for metabolic diseases?  A multivariate method such as PLS-DA, Random Forest, or PCA may yield some interesting and useful results.  These are conveniently implemented using the statistical packages available at: https://www.metaboanalyst.ca.

Response to point 1. We thank the reviewer for this suggestion. We are familiar with the use of MetaboAnalyst and we also performed PCA, PLS-DA and random forest analyses using R programming language. Since the sample size for each individual IEM is rather small, a supervised clustering analysis performed with a lot of mass peaks is unreliable, as it will always be able to distinguish the groups. However, it was reassuring to find that mass peaks with the highest VIP-scores were the mass peaks annotated with the metabolites that are demonstrated in Figure 2. Using unsupervised clustering analyses, we were indeed able to see some clustering of control samples versus specific IEM, however, due to the small sample size per IEM we decided that clustering analyses and biomarker identification should not be the scope of this study. We will perform further (clustering) studies for biomarker identification, by analyzing larger sample sizes – if available – for specific IEM.

Point 2. The reproducibility of the method was assessed using 20 isotope standards. Twenty compounds are too few to represent the total 1875 identified compounds and 3835 isomers in the samples. Most of the 20 isotopes are amino acids and acylcarnitines. These compounds are usually abundant in plasma and DBS. How about the RSD of compounds with low abundance in the samples?  For example, orotic acid, a key biomarker for OTC deficiency? How about methylmalonic acid for Vitamin B12 deficiency or methylmalonic acidemia? More diverse representation of the major chemical classes should be selected as isotope standards to evaluate the reproducibility of the method in future implementations of these methods.

Response to point 2. We thank the reviewer for this valid notion. We agree that only a few major chemical classes are represented in the selection of stable isotope-labeled compounds. We use these compounds to assess within-batch reproducibility of the results as a whole, and not specifically to assess reproducibility of acylcarnitines and amino acids. Unfortunately it is not feasible to add stable isotope-labeled compounds for all annotated metabolites that can occur endogenously.

For reproducibility of data interpretation, we use the between-batch reproducibility of the Z-scores of positive control samples. This was assessed in three patients, in seven batches. In the manuscript we demonstrated the RSDs of seven compounds (Table 3), but we assessed the data that is available for all compounds. For example, the organic acid 2-Amino-3-phosphonopropionic acid was also increased in the PA patient sample, and Z-scores were stable: 2.89, 3.29, 2.77, 2.73, 4.73, 3.94 and 2.91, RSD 0.23. Our claim that calculated Z-scores are fairly stable also holds true for methylmalonic acid, which is in the normal range in the PA patient sample: -0.64, -0.66, 0.51, 0.93, -0.83, 0.29 and -0.70 and for orotic acid, also in the normal range in this patient sample: 0.82, -1.00, -0.28, -0.92, -0.21, 0.29 and -0.67. Other examples are vanillylmandelic acid, increased in the PKU patient sample: 9.26, 8.22, 5.35, 12.08, 12.57, 15.17 and 5.57, RSD 0.38 and guanidoacetic acid, increased in the LPI patient sample: 6.02, 4.45, 4.70, 6.66, 3.91, 6.20 and 3.10, RSD 0.26.

This data demonstrates that reproducibility of metabolites in other major chemical classes such as organic acids, is also satisfactory. Since these analyses provide reproducibility data for all annotated compounds, we are confident that for within-batch reproducibility addition of twenty stable isotope-labeled compounds is sufficient. To convey this message, we added the sentence “Many other compounds annotated in these three patients, representing major chemical classes of metabolites as acylcarnitines, amino acids and organic acids, were found to have satisfactory RSDs (all data is available on request).” to section 2.1 of the results.

Point 3. In Table 4, the authors showed the Z-score of methylmalonic acid (MMA).  How did you accurately identify MMA? How do you avoid the interference of the atomically identical (C4H6O4) and structurally related isomer succinic acid? Both compounds have a monoisotopic mass of 118.026611 Daltons. Succinic acid is usually present in samples at a greater concentration than MMA.

Response to point 3. The reviewer here addresses one of the key characteristics of the DI-HRMS method. Indeed, using DI-HRMS metabolomics, it is not possible to distinguish isomers. Therefore, we cannot distinguish methylmalonic acid from succinic acid. We annotated both methylmalonic acid and succinic acid to the mass peak of 118.026611 Daltons. The subscript of Table 4 declares that for each mass peak, only one metabolite annotation is reported, the one that influenced the final decision. Since propionylcarnitine is clearly increased in these patient samples (Z-scores 13.3 and 75.4), as well as 2-methylcitric acid (7.3 and 4.3), our differential diagnosis included both propionic acidemia and methylmalonic acidemia. The increase of a mass peak that is annotated with (among others) methylmalonic acid (16.6 in patient 2) led to a most probable diagnosis of methylmalonic aciduria. Though, we do not state that we identified the metabolite methylmalonic acid in this patient, we demonstrate that we are able to characterize the patient biochemical profile. We stated in the discussion: “Semi-quantitative metabolomics always produces tentative results. When used in daily practice, we advocate confirmation of the results with second-tier testing, using targeted diagnostic platforms or genetic tests.”

Point 4. Subjects’ metadata (age, sex, etc) is missing for this study. These metadata are essential for other researchers to use this method and compare the results across the studies.

Response to point 4. Indeed, we agree to the reviewer that age and sex are relevant as it can influence an individuals’ metabolome. However, also time of day at sampling, drugs, supplements, diets, storage time of the sample and degree of hemolysis could influence the metabolome of an individual at a certain point in time. Unfortunately, it is impossible to correct for all of this, or to match patient and control samples on all these factors. Still, these factors should be taken into account. To do this, we compared all patient samples to a group of 30 control samples, varying in all these factors. By taking into account the standard deviation of the control group when calculating Z-scores, a Z-score indicating aberration from the mean of > 2 SD is quite likely to be biologically relevant for the inborn error of metabolism, more than all other possibly influencing factors. To make this more clear in our manuscript, we changed the manuscript accordingly by adding the following sentence to section 4.2: “Control samples varied in age (including both neonates and elderly), sex, time of day at sampling, diets, supplements, drugs, storage time and for control plasma samples degree of haemolysis, in order to take into account considerable variation in factors that might influence an individuals’ metabolome.”

Point 5. In the method section (4.1), DBS were prepared either by spotting the blood taken from heparin-containing tubes or by the finger prick. The concentration of certain metabolites is different between these 2 approaches. In the future, a more standardized collection protocol should be used.

Response to point 5. We completely agree to the reviewer on this point, ideally a more standardized collection procedure should be used. We normally prepare DBS by spotting the blood taken from heparin-containing tubes. However, for studies like this the limited availability of positive control samples is always a challenge and since a few samples collected by finger prick were available, we did not want to exclude these samples for this study. In future studies, especially when focusing on biomarker identification, we will apply a more standardized sample collection procedure.
